# A “Trojan Horse” Strategy: The Preparation of Bile Acid-Modifying Irinotecan Hydrochloride Nanoliposomes for Liver-Targeted Anticancer Drug Delivery System Study

**DOI:** 10.3390/molecules28041577

**Published:** 2023-02-07

**Authors:** Tao Zhou, Yushi Liu, Kelu Lei, Junjing Liu, Minghao Hu, Li Guo, Yiping Guo, Qiang Ye

**Affiliations:** 1State Key Laboratory of Southwestern Chinese Medicine Resources, Chengdu University of Traditional Chinese Medicine, Chengdu 611137, China; 2College of Pharmacy, School of Modern Chinese Medicine Industry, Chengdu University of Traditional Chinese Medicine, Chengdu 611137, China

**Keywords:** “Trojan Horse” Strategy, irinotecan hydrochloride, cholic acid, liposomes, targeted modification

## Abstract

The bile acid transport system is a natural physiological cycling process between the liver and the small intestine, occurring approximately 6–15 times during the day. There are various bile acid transporter proteins on hepatocytes that specifically recognize bile acids for transport. Therefore, in this paper, a novel liposome, cholic acid-modified irinotecan hydrochloride liposomes (named CA-CPT-11-Lip), was prepared based on the “Trojan horse” strategy. The liposomes preparation process was optimized, and some important quality indicators were investigated. The distribution of irinotecan hydrochloride in mice was then analyzed by high-performance liquid chromatography (HPLC), and the toxicity of liposomes to hepatocellular carcinoma cells (HepG-2) was evaluated in vitro. As a result, CA-CPT-11-Lip was successfully prepared. It was spherical with a particle size of 154.16 ± 4.92 nm, and the drug loading and encapsulation efficiency were 3.72 ± 0.04% and 82.04 ± 1.38%, respectively. Compared with the conventional liposomes (without cholic acid modification, named CPT-11-Lip), CA-CPT-11-Lip had a smaller particle size and higher encapsulation efficiency, and the drug accumulation in the liver was more efficient, enhancing the anti-hepatocellular carcinoma activity of irinotecan hydrochloride. The novel nanoliposome modified by cholic acid may help to expand the application of irinotecan hydrochloride in the treatment of hepatocellular carcinoma and construct the drug delivery system mode of drug liver targeting.

## 1. Introduction

Liver cancer is a malignant tumor, which seriously threatens human life and health. As the pathogenesis of liver cancer is not completely clear, surgery is still the main treatment method. However, postoperative patient satisfaction and survival rates are relatively poor [1]. Chemotherapy is another common treatment for cancer because of its high efficacy [2]. Nevertheless, patients may suffer many side effects due to the disadvantages of chemotherapeutic drugs, such as inferior physical and chemical properties, low bioavailability, and inaccuracy tissue selectivity [3]. Therefore, it is imperative to find safer and more effective natural products for cancer treatment. Irinotecan hydrochloride (CPT-11) is a semisynthetic water-soluble derivative of camptothecin extracted from *Camptotheca acuminata Decne*. It has been shown to have stronger antitumor activity and less toxicity than camptothecin and is a specific inhibitor of type I DNA topoisomerase [4]. As an anticancer drug, CPT-11 exhibits strong anticancer activity against a variety of cancers, such as colorectal cancer, liver cancer, pancreatic cancer, gastric cancer, cellular lung cancer, cervical cancer, ovarian cancer, etc. [5,6].

Although CPT-11 is effective, it still has many disadvantages, such as short half-life, delayed diarrhea, and neutropenia, etc., which are recognized as constituting the dose-limiting toxicity for this drug [7,8,9,10]. Additionally, it should be avoided, since the lactone configuration of CPT-11 is inactivated in vivo by hydrolysis to an inactive carboxylate [11]. Therefore, an effective drug delivery system to reduce toxicity and maintain the active form of the drug is urgently desired. A major difficulty facing cancer drug delivery today is the insufficient uptake and accumulation of drugs in cancer cells [12]. Modern research focuses on finding effective drug delivery systems that enable the decrease of administration with the purpose of reducing drug toxicity [13,14]. Nanoparticles play an important role in cancer therapy, such as Ansari et al.’s prepared hispolon-loaded liquid crystalline nanoparticles, which significantly improved the biological indices associated with liver cancer [15]; Shnoudeh et al. fabricated gold (Au), iron (Fe), and selenium (Se) nanoparticles (NPs) with an excellent biocompatibility [16]. Liposomes are widely studied nanoparticles, which are lipid bilayers of bubbles encapsulating drugs. The phospholipid bilayer is similar in properties and functions to biological membranes, which can fuse with cell membranes to deliver drugs into the interior of cells. In addition, cholesterol-rich liposomes activate the human complement system and improve adverse hemodynamic changes [17]. Compared with normal drugs, liposomes can enhance permeability and retention effects (EPR), prolong circulation time, and achieve more drug accumulation in the tumor region [18]. For liposomes, physicochemical aspects including size distribution, shape, rigidity or deformability, and surface charge determine the manner of entry into the cell, intracellular transport, release of the contents, and toxicity of the nanoparticles [19]. Therefore, improvement of the properties of normal liposomes is necessary. Another concern is that the passive targeting of traditional liposomes cannot ensure increased the uptake of drugs by tumor cells to fully exploit the efficacy. In recent years, scholars have paid much attention to modifying the lipid membrane surface to improve active targeting. For example, Zhu et al. found that ginsenosides combined with paclitaxel liposomes were superior to most reported paclitaxel preparations, in which ginsenosides played the dual role of drug and adjuvant and successfully targeted the drug to gastric tumors [20].

Furthermore, research studies have shown that many drugs’ transmembrane transport is mediated by specific membrane transport proteins, rather than simple diffusion [21]. Transport proteins on the cell membrane can specifically recognize certain drugs to increase the drug’s absorption. For molecular drug delivery, membrane transporters are so important in drug pharmacokinetics, safety, and pharmacodynamics behavior that they are regarded as “Trojan horses” [22]. Tumor cells often invade some important organs, such as liver, lung, bone tissue, etc., causing lesions in these organs. Meanwhile, these occupied organs and tumor cells are like “seed and soil”, providing nutrition and protection for tumor cells [23]. The diseased organs build hard castles for the tumor cells, and in order to kill the tumor cells, the drug needs to enter these castles (organs) firstly, which requires modification of the drug’s surface for better integration with specific receptors on the target organs. The “Trojan horse” strategy is to utilize the specifically recognitions of the transporter on the cell membrane to the compounds with certain specific structure, then combine the compound with other drugs to enhance the absorption of the drug by active targeting [22,24]. “Trojan horse” strategies have been widely applied in nanoparticles. Chitosan nanoparticles loaded with drugs have stability, permeability, and bioactivity [25]. Guo et al. used modified nanoparticles loaded with a cell-penetrating peptide and amphipathic chitosan derivative as carriers to improve colonic absorption of the drug [26]. Soe et al. used the high expression of folate receptors in colorectal cancer cells to modify the nanostructured chitosan/chondroitin sulfate composite carrier with folate to enhance the transportation of bortezomib to colorectal cancer cells [27]. Kucharz et al. attached antibodies against the transferrin receptor to the surface of nanoparticles encapsulating cisplatin, allowing for the drug-loaded nanoparticles to successfully cross the blood–brain barrier and achieve transmembrane transport of the drug to the brain [28]. Zhou et al. have successfully targeted drugs to the liver using searyl glycyrrhetinate-modified liposomes by exploiting the specific recognition of searyl glycyrrhetinate by the glycyrrhetinic acid receptor [29]. In conclusion, ligand-modified nanoparticles can actively accumulate in tumors through a ligand–receptor combination mechanism, thereby delivering drugs to the targeted regions. Therefore, finding the relevant membrane transporter proteins is the critical for drug-loaded nanoparticle-targeted delivery.

There are various bile acid transporters, such as apical sodium-dependent bile acid transporter (ASBT), Na^+^-taurocholate co-transporting polypeptide (NTCP), and organic anion transporting polypeptide (OATP), in intestinal cells and hepatocytes [30]. Bile acid transporters are highly expressed in intestinal cells and hepatocytes and appear to be ideal transporters for a “Trojan horse” strategy. Among bile acids, cholic acid (CA) is a specific natural ligand for endogenous cells and has a high organ affinity. At the same time, CA as a cholesterol derivative often plays an adjuvant role in liposomes [31]. Therefore, in this study, based on the “Trojan horse” strategy, the CA was selected as the natural ligand and CA-modified CPT-11 liposomes were prepared. CPT-11-Lip and CA-CPT-11-Lip were prepared by the film dispersion method. The preparation process of CA-CPT-11-Lip was optimized by the Box-Behnken experimental design (BBD) method, and the characteristics of the optimized liposomes, such as encapsulation efficiency (EE), drug loading (DL), particle size, zeta potential (ZP), polydispersity index (PDI), stability, and in vitro release, were evaluated. After that, the tissue distribution in mice was comparatively studied to observe the accumulation of drugs in different tissue. Finally, the in vitro anticancer activity of two liposomes on hepatoma cells (HepG-2) was evaluated.

## 2. Results

### 2.1. The Results of BBD Study

Taking the EE and DL as evaluation indicators, the BBD experiment was designed with three main factors: lecithin dosage, lecithin–cholesterol ratio, and drug dosage. The relevant data for optimization are shown in Table 1, Table 2, Table 3 and Table 4, and the 3D response surface plots are shown in Figure 1. The results were analyzed by ANOVA using Design-Expert 12 software, and the regression equations of EE (R1) and DL (R2) were as follows:R1 = 64.57 + 16.38A + 9.79B − 14.88C − 4.78AB − 0.75AC + 0.455BC − 13.33A2 − 7.55B2 + 1.91C2
R2 = 4.49 − 024A + 1.02B + 0.75C − 0.72AB + 0.36AC + 0.455BC − 0.8315A2 − 0.4365B2 − 0.5065C2

The results of the one-way ANOVA for the EE (Table 3) and DL (Table 4) are shown that the model *p* < 0.01 and the lack of fit item *p* > 0.05, indicating that the model can be used for optimal process prediction and has applicable value. The quadratic equation and 3D response-surface morphology reflect the effects of independent variables on liposome EE and DL [32]. In 3D effect surface map, the variations in color and the degree of tilt reflected the effects of independent factors on encapsulation rate and drug loading, and usually the darker the color and the larger the slope, the more significant the influence. From the quadratic equations R1 and R2 and Figure 1, we can find that factors A and C had a significant effect on EE, and there was no interaction between factors A, B, and C. The EE tended to improve with increasing lecithin or decreasing CPT-11, as they both enhanced the chance of drug encapsulation inside the liposomes. In addition, the effect of independent factors on DL is complicated. Factors A and B had a strong influence on DL, but all three factors interacted with each other.

Usually, the EE of liposomes is required to be more than 80%. Meanwhile, the level of DL directly affects the clinical application dose of the drug, and the higher the DL, the more adapted to clinical needs. Therefore, the optimal preparation process of CA-CPT-11-Lip was predicted with a weighting ratio of 3:1 for EE and DL: 104.80 mg of lecithin, 6.28:1 ratio of lecithin-cholesterol, and 6.45 mg of CPT-11. Meanwhile, EE reached a forecast of 80.27% and DL reached 3.92%. As shown in Table 5, the deviations between the measured and predicted values of the EE and DL were 1.77% and −0.2%, respectively, indicating that the results of the binomial fitting were good, and the reliability was high.

### 2.2. Characterization of the CPT-11-Lip and CA-CPT-11-Lip

#### 2.2.1. Morphology, EE, DL, Particle Size, Zeta Potential, and PDI

As shown in Figure 2, the morphology of the two liposomes was spherical, and the particle size was in accordance with our determination. Some other characteristics are shown in Table 6. The zeta potential values of CA-CPT-11-Lip and CPT-11-Lip were −56.93 ± 0.46 and −53.07 ± 1.47, respectively. Usually, a higher zeta potential is associated with better stability of the formulation, so both liposomes had certain stability [33]. The particle sizes of both liposomes were about 154.16 ± 4.92 nm and 197.70 ± 3.04 nm, respectively, and the PDI was small (PDI < 0.25). The particle size was less than 200 nm, which is beneficial to avoid physical clearance [29]. The EE of CPT-11-Lip was 74.54% and the DL was 3.73%, while CA-CPT-11-Lip was 82.04% and 3.72%. In terms of particle size, PDI and EE, CA-CPT-11-Lip showed advantages. Its particle size was smaller, PDI was lower, and EE was higher, especially the difference in particle size was obvious (Figure 3). Therefore, CA optimizes the properties of conventional liposomes.

#### 2.2.2. Particle Size Stability

The change in particle size was used to evaluate the stability of the liposomes. As shown in Figure 4, there was no significant changes in the 15 days. In conclusion, the appearances of both liposomes were uniform within 15 days, and the particle size was constant, indicating that the preparation could be stored stably at 4 °C for 15 days. Notably, the liposomes were precipitated after 20 days. The stability of liposomes is influenced by several factors, which often limit their application [34]. Therefore, it is necessary to explore how to further improve the stability of the formulation in the future, such as considering the preparation of liposomes into lyophilized formulations [35].

#### 2.2.3. In Vitro Release

The drug release curves of CPT-11 solution, CPT-11-Lip, and CA-CPT-11-Lip are shown in Figure 5. The CPT-11 solution group was rapidly released in the PBS 7.0 solution, with the cumulative release rate reached approximately 80% within 6 h. However, the release rate of CPT-11-Lip group and CA-CPT-11-Lip group only reached 40% after 12 h, which reflected the sustained release effect of liposomes. The release profiles of liposomes groups were similar, indicating that the addition of CA did not affect the release in vitro. In addition, compared with PBS 7.0 group, the release rate of liposomes in PBS 5.0 solution was significantly increased, especially in the CA-CPT-11-Lip group, which could reach 60% at 12 h. This might be that the acidic environment accelerated the release of CPT-11 from liposomes.

### 2.3. The Results of Tissue Distribution Study

The distribution results of the CPT-11 in different tissues of mice after gavage administration are shown in Figure 6. Overall, the three preparations, CPT-11 solution, CPT-11-Lip, and CA-CPT-11-Lip, were most distributed in the liver, followed by the spleen, kidneys, and lungs, and they were the least distributed in the heart. After 1 or 2 h of administration, CPT-11 reached high concentrations in every tissue, and after 4 h, the drug was almost undetectable in all tissues. As shown in Figure 6A, CPT-11 solution reached a high concentration in the liver at 0.5 h and reached its maximum concentration at 1 h, after which it was rapidly cleared. CPT-11-Lip was less distributed at 0.5 h, reached its maximum concentration at 1 h, and then slowly decreased. However, CA-CPT-11-Lip reached a high concentration at 1 h and a maximum concentration at 2 h, after which it remained the highest concentration among the three preparations. These indicated that CA-CPT-11-Lip accumulated in the liver and was slowly released. As shown in Figure 6F, considering the total amount of CPT-11 in the five tissues as 100%, we calculated the percentage content of the CPT-11 in the liver at different time for better analysis. It can be seen that the distribution percentages of CA-CPT-11-Lip in the liver were the highest and exceeded 50%. Interestingly, the difference between the two liposomes at 3 or 4 h was significant (*p* < 0.01), further illustrating the liver-targeting and slow-release effects of CA-CPT-11-Lip.

### 2.4. In Vitro Cytotoxicity Study

The cytotoxicity of CA, CPT-11, CA + CPT-11 (physical mixture of CA and CPT-11), C-Lip (blank liposomes), CPT-11-Lip, and CA-CPT-11-Lip for HepG-2 hepatocellular carcinoma was evaluated by CCK-8 method. The effect of each formulation on HepG-2 cell survival at 50 μM concentration is shown in Figure 7A. It can be seen that the C-Lip group promoted cell growth with a proliferative effect; the cell survival rate of the CA group was higher than 80%, and had almost no inhibitory effect on the proliferation of hepatocellular carcinoma cells; the HepG-2 cell survival rate of CPT-11 and CA + CPT-11 groups was 30% and showed cytotoxicity of hepatocellular carcinoma; meanwhile, the survival rate of CPT-11-Lip and CA-CPT-11-LIP groups was about 10%, which showed a significant inhibitory effect. There was no significant difference (*p* > 0.05) between the inhibition of the two groups of liposomes, but they both showed better anticancer activity than CPT-11 groups (*p* < 0.01). Thus, liposomes enhanced the toxicity of CPT-11 on hepatocellular carcinoma cells.

Figure 7B showed the effects of different concentrations of CPT-11, CPT-11-Lip, and CA-CPT-11-Lip on the survival of HepG2 cells. The three formulations showed the same inhibition curves, and the cell viability decreased with the increase of CPT-11 concentration, indicating that the inhibitory effect of CPT-11 on HepG-2 cells was dose-dependent. The IC50 values of CPT-11, CPT-11-Lip, and CA-CPT-11-Lip on HepG-2 cells were 15.21 ± 0.47, 6.86 ± 0.66, and 6.02 ± 080, respectively. The IC50 values of the two liposomes were smaller than that of the CPT-11 group, which further illustrated the superiority of the liposomes drug delivery system.

### 2.5. Effects of Cholic Acid on Cell Uptake of CPT-11

As shown in Figure 8, cholic acid pretreatment significantly attenuated the uptake of CA-CPT-11-Lip by HepG2 cells in a dose-dependent manner. In contrast, there was no significant change in the CPT-11-Lip and CPT-11 solution groups. These results were consistent with the study of Xiao et al., suggesting that cholic acid-modified liposomes can increase drug uptake, possibly through bile acid transport proteins on hepatocellular carcinoma cells [36]. In addition, we found that drug uptake was higher in the liposome groups than in the CPT-11 group, and the value was greatest in the CA-CPT-11-Lip group, further illustrating the liver targeting of cholic acid-modified liposomes.

## 3. Discussion

In this study, CA was selected to modify CPT-11 nanoliposomes, and CA-CPT-11-Lip was successfully guided with CPT-11 into the liver. The liposomes delivery system showed to have a better effect on liver cancer cells than CPT-11. In the experiment, we obtained some striking results.

### 3.1. Choice of Liposomes Preparation Method

Generally, active drug loading techniques are considered more suitable for the preparation of weakly alkaline drug liposomes [37,38], although they are complicated, tedious, and not appropriate for large-scale industrial production. It also seems to be the optimal preparation method for CPT-11 liposomes [39]. In the BBD process optimization (Table 2), we found that the preparation of CA-modified liposomes by the film dispersion method was rather simple and reproducible. This meant that CA as a membrane material simplified the preparation of CPT-11 liposomes.

### 3.2. Optimization of EE, Particle Size and PDI

From Table 6, CA improved the EE of the liposomes and reduced the particle size and PDI. The particle size of liposomes largely determines their performance in vivo. Usually, reducing the particle size increases the diffusion ability of liposomes, which benefits the diffusion to the tumor area and enhances the uptake of tumor cells for therapeutic effect [40]. PDI is a parameter describing the homogeneity of liposomes particle size. The uniform particle size liposomes can release the drug at the site of action homogeneously [41]. Notably, the accuracy of EE is the key to this experiment. We found that the flow rate affected the calculation of EE when using a Sephadex G-50 gel column to separate free drug and liposomes. Typically, liposomes caused gel clogging during separation, slowing down the flow rate. Therefore, the gel column should be cleaned with large eluent after usage. In severe cases, it was necessary to regenerate the gel.

### 3.3. Targeting and Effects

The most meaningful aspect of this study was the tendency of CA-modified liposomes to accumulate consistently in the liver. As shown in Figure 6F, CA-modified liposomes allowed for CPT-11 to maintain high concentrations in the liver for 4 h, which provided a physical basis for CPT-11 to treat liver disease. Many studies have been performed using bile acids to target drug transport to the liver, demonstrating that the “Trojan horse” strategy of bile acids is well-established and effective [22]. However, the majority of these have focused on the conjugates of bile acids but did not change the shortcomings of the chemotherapeutic drugs themselves. In this study, the combination between the “Trojan horse” strategy of bile acids and liposomes not only efficiently transported CPT-11 to the liver, but also protected the drug structure from being damaged and exerted its efficacy at specific sites. Additionally, in the cytotoxicity and cellular uptake experiments in vitro, the IC_50_ of CA-CPT-11-Lip was smaller than CPT-11, and the intracellular uptake was increased, which demonstrated the advantages of CA-CPT-11-Lip.

### 3.4. Potential of CA as a Liposomal Membrane Material

The properties of CA were used to infer its role on liposomes; it is a sterol derivative with a structure similar to cholesterol [42]. Therefore, CA can function as a membrane material and affect the rigidity and stability of liposomes together with cholesterol. Moreover, CA as well as lecithin possess amphiphilicity and can be ordered in solution, which provides the possibility of CA modification of liposomes [43]. Besides, bile acids are important endogenous substances in the enterohepatic circulation of the body, so they have better integration with cell membranes [44]. It has also been reported that 70% lipid and 30% cholesterol is the most ideal formulation combination [45]. In the BBD study, the low lecithin-cholesterol ratio group (3:1) was found to have a low EE, while the high group (7:1) had a high EE. In other words, the group close to the ideal formulation did not obtain a higher EE. This may be due to CA rebalancing the “lecithin-cholesterol ratio”, which revealed the function of CA as an adjuvant. These results suggested that the “Trojan horse” strategy was successful, allowing for the efficient transportation of CPT-11 to the liver, which can improve the therapeutic effects of CPT-11 and reduce its toxic effects.

## 4. Materials and Methods

### 4.1. Materials, Cell Lines, and Animals

Irinotecan hydrochloride (CPT-11, purity > 99%, Chengdu Refensi Biotechnology Co., Ltd., Chengdu, China), cholic acid (purity ≥ 98%, Chengdu McLean Biotechnology Co., Ltd., Chengdu, China), lecithin (purity ≥ 99%, Chengdu Kelon Chemical Co., Ltd., Chengdu, China), and cholesterol (purity > 95%, Chengdu Kelon Chemical Co., Ltd., Chengdu, China) were obtained. Acetonitrile was HPLC grade. All other reagents were analytical grade, and purified deionized water was used throughout.

HepG2 cell lines were provided by the State Key Laboratory of Southwestern Chinese Medicine Resources. Cells were cultured in DMEM medium containing 10% fetal bovine serum and 1% penicillin-streptomycin at 37 °C in a 5% CO_2_ constant temperature incubator.

KM mice (18–22 g) were purchased from Spearfish (Beijing) Biotechnology Co., Ltd. (Beijing, China). All mice were acclimatized to their new environment for one week, during which they were kept in light for 12 h and dark for 12 h per day, with appropriate control of diet and free access to water. Mice were prohibited from ingesting food for 12 h before the experiment, but drinking water was not restricted. All animal experimental protocols were approved by the Animal Research Ethics Committee of Chengdu University of Traditional Chinese Medicine (NO.2022-29).

### 4.2. Preparation of CPT-11-Lip and CA-CPT-11-Lip

Liposomes were prepared by film dispersion method. First, 100 mg of lecithin and 20 mg of cholesterol were weighed into a 250 mL eggplant-shaped bottle, and 20 mL of a chloroform–methanol (9:1) mixed solution was added for sonication and placed on a rotary evaporator (RE-52AA, ShangHai Yarong Biochemistry Instrument Factory, Shanghai, China) at 40 °C to remove the organic solvent under vacuum until a uniform film formed on the bottle wall. Then, 10 mL of 1.0 mg/mL CPT-11 aqueous solution was added to the lipid film and hydrated for 1.5 h. The formed liposomes’ solution (CPT-11-Lip) was then sonicated for 5 min in an ice-water bath using an ultrasonicator (KM-500DE, Kunshan Meimei Ultrasonic Instrument Co., Ltd., Kunshan, China) to disperse it uniformly, and then filtered through 0.45 and 0.22 μm microporous membranes sequentially and stored at 4 °C. CA-CPT-11-Lip was prepared in the same manner with an additive of 15 mg of CA in the 250 mL eggplant-shaped bottle.

### 4.3. BBD Study

According to the literature survey, the dosage of lecithin, lecithin–cholesterol dosage ratio, and the dosage of CPT-11 were designed and optimized using Box-Behnken 12 software [46]. The quadratic model was established by the three-factor BBD method, and each factor was divided into three levels: high (1), medium (0), and low (−1). Analysis of variance (ANOVA) was further used on the experimental results to analyze the statistically significant terms of the quadratic model. Finally, we used the data from the quadratic model to create 3D response surface plots.

### 4.4. Characterization of the Liposomes Study

#### 4.4.1. Drug Loading (DL) and Encapsulation Efficiency (EE)

According to the BBD results, CPT-11-Lip and CA-CPT-11-Lip (*n* = 3) were prepared under the optimal process. Free-formed drug and liposomes were separated using a Sephadex G-50 gel chromatography method [29]. The inner diameter of the chromatographic column was 2.5 cm. The loading height of the dextran gel G-50 was about 15 cm. The sample volume of liposome was 1 mL, and pure water was used as the elution solvent with a flow rate of 0.5 mL/min. Formula (1) and (2) were used to calculate the EE and DL of liposomes, respectively.
(1)$$\mathrm{EE}\%=\frac{\;\mathrm{encapsulated}\;\mathrm{drug}\;\mathrm{content}}{\mathrm{total}\;\mathrm{drug}\;\mathrm{content}}\times 100\%$$
(2)$$\mathrm{DL}\%=\frac{\mathrm{encapsulated}\;\mathrm{drug}\;\mathrm{content}}{\mathrm{weight}\;\mathrm{of}\;\mathrm{carrier}}\times 100\%$$

#### 4.4.2. Morphology, Particle Size, Zeta Potential, Polydispersity Index (PDI), and Stability In Vitro

The morphology of the liposomes was observed and photographed by transmission electron microscopy (JEM-1400FLASH, JEOL, Tokyo, Japan). After the sample was properly diluted, an appropriate amount was dropped on the copper mesh. Before further analysis, it was stained with 2% phosphotungstic acid at room temperature for 1–2 min, and then observed and photographed. The particle size, zeta potential, and PDI of liposomes were determined by a nanoparticle size and Zeta potential analyzer (Litesizer 500, Anton Paar, Graz, Austria). Meanwhile, the particle size of CPT-11-Lip and CA-CPT-11-Lip were measured at 1, 3, 7, and 15 days, respectively, to analyze and evaluate the stability of liposomes.

#### 4.4.3. In Vitro Drug Release Study

The cumulative release of CPT-11 solution, CPT-11-Lip, and CA-CPT-11-Lip in vitro was evaluated by dialysis method. First, 0.5 mL of CPT-11 solution, CPT-11-Lip, and CA-CPT-11-Lip, ensuring similar levels of CPT-11, were taken into a dialysis bag with a molecular weight cutoff of 8–12 kDa. Next, 10 mL of PBS (pH value is 5.0 or 7.0, respectively) was added as dialysis medium and dialyzed at room temperature at 100 rpm on a shaker (SLK-O3000-S, SCILOGEX, America) at room temperature. Then, 1 mL of medium was taken out at 0, 0.5, 1, 2, 4, 8, 12, 24, 36, and 48 h, respectively, and an equal amount of medium was supplemented at the same time. After the sample was passed through a 0.22 um microporous membrane, the content was determined using a high-performance liquid spectrometer (1260 infinity II, Agilent, Palo Alto, CA, America). The cumulative release of the drug was calculated according to Formula (3), in which W_release_ is the total amount of CPT-11 released at a specific time and W_total_ is the total amount of CPT-11 in the initial dialysis bag.
(3)$$\mathrm{Cumulative}\;\mathrm{drug}\;\mathrm{release}\%=\frac{W\;\mathrm{release}}{W\;\mathrm{total}}\times 100\%$$

### 4.5. Tissue Distribution Study

KM mice were randomly divided into a blank group and an administration group (the CPT-11, CPT-11-Lip, and CA-CPT-11-Lip groups). In the administration group, CPT-11, CPT-11-Lip, and CA-CPT-11-Lip were administered to mice by gavage at a dose of 50 mg/Kg [47], respectively. Animals were sacrificed at 0.5, 1, 2, 3, 4, 8, and 12 h after administration and the heart, liver, spleen, lungs, and kidneys were gathered. The tissue samples were weighed and added with 2 times the saline; the samples were homogenized with a homogenizer (Tissuelyser-48, Shanghai Jingxin Technology, Shanghai, China). Then, the tissue solution was centrifuged at 4 °C and 10,000 r/min for 10 min, the supernatant was collected, an equal amount of acetonitrile was added to completely precipitate the protein, and the supernatant was collected by centrifugation again and stored at −80 °C. The CPT-11 content was detected at 370 nm using a reversed-phase column (Zhongpu Science, 4.6 × 250 mm, 5 μm). The mobile phase was a mixture of acetonitrile-0.1% trifluoroacetic acid aqueous solution (from 0 to 15 min, the volume of acetonitrile was increased from 20% to 80%) at a flow rate of 1 mL/min and an injection volume of 20 μL. Standard tissue samples were prepared from blank tissue samples and CPT-11 standard solution, and a standard curve was established to calculate the concentration of CPT-11 in each tissue.

### 4.6. In Vitro Cytotoxicity Study

The in vitro toxicity of liposomes to hepatoma cells (HepG-2) was evaluated by the CCK-8 method. The log-phase HepG-2 cells with good growth status were seeded in 96-well plates at 4 × 103 cells per well. After incubation for 24 h, appropriate concentrations of CPT-11, CPT-11-Lip, and CA-CPT-11-Lip were added, respectively. Sampled were then incubated for another 48 h, then we discarded the culture medium, added 10 μL/well of CCK-8 solution in the dark, and continued to incubate for 1.5 h. Finally, using a Multifunctional Enzyme Analyzer (SpectraMax iD, Thermo Scientific, Waltham, MA, USA), we detected the absorbance at the wavelength of 450 nm. The cell viability was calculated according to Formula (4):(4)$$\mathrm{Cell}\;\mathrm{viability}\;\left(\%\right)=\frac{\mathrm{OD}\;\mathrm{experimental}\;\mathrm{group}}{\mathrm{OD}\;\mathrm{blank}\;\mathrm{group}}\times 100\%$$

### 4.7. CPT-11 In Vitro Uptake Study

According to the method of Xiao et al. to verify the effect of cholic acid on cellular uptake. [36], HepG2 cells in the logarithmic growth phase were inoculated by aspirating 1 mL in 12-well plates at a concentration of 1 × 10^4^ cells per well and placing them in an incubator. After 24 h, the culture medium was pipetted out, washed once with PBS, and pretreated for 0.5 h by adding different concentrations of CA. The CA solution was then aspirated, and the cells were washed with PBS. Then, 10 μM CPT-11 solution, CPT-11-Lip, and CA-CPT-11-Lip were added and incubated for 2 h. The cells were then washed with PBS and single-cell suspensions were collected afterwards. Next, cells were washed with cold PBS, and 250 μL PBS was added to resuspend the cells. The cell suspensions were freeze-thawed three times, then sonicated for 10 min, followed by centrifugation at 10,000 rpm for 10 min, and the supernatant was taken to determine CPT-11 and protein concentrations. To determine the concentration of CPT-11, 200 μL of supernatant was added to an equal volume of acetonitrile to precipitate the protein, vortexed for 2 min, centrifuged, and the supernatant was collected, and the absorbance was measured at 371 nm by an enzyme standardization instrument. The concentration of the compound was calculated by the corresponding standard curve. The protein concentration was determined according to the protein quantification (TP) assay kit (Nanjing Jiancheng Institute of Biological Engineering, Nanjing, China). The intracellular CPT-11 uptake was expressed as the amount of CPT-11 associated with 1 mg of cellular protein.

### 4.8. Statistical Analysis

All experiments were performed at least three times in parallel, and the experimental data were expressed as mean ± standard deviation (SD). An unpaired t-test was performed on the data using SPSS 26.0 software, and *p* < 0.05 was considered statistically significant.

## 5. Conclusions

In this study, CPT-11-Lip and CA-CPT-11-Lip were successfully prepared by the film dispersion method, and the process parameters were optimized by the BBD method. The study found that CPT-11-Lip and CA-CPT-11-Lip were similar in zeta potential, PDI, and drug loading, indicating that CA had no significant effect on the physical properties of liposomes. However, CA-CPT-11-Lip has a smaller particle size of 154.16 ± 4.92 and a higher EE of 82.04 ± 1.38%, which showed the advantages of CA as an excipient. In addition, liposomes exhibited sustained release in in vitro cumulative release studies. In tissue distribution studies, the liposomes group exhibited drug accumulation and sustained release in the liver, while the CA-CPT-11-Lip group showed more significant effects, confirming our proposed “Trojan horse” strategy. Finally, in the liver cancer cytotoxicity test, the liposomes’ effect was more significant compared with the CPT-11 group, reducing the IC50 value by nearly half. These indicated that CA was a promising excipient for liposomes, and more formulation studies should focus on it in the future. In conclusion, CA-CPT-11-Lip can deliver CPT-11 to the liver via bile acid receptors and fully exploit the therapeutic effect, which provides a reference for the clinical application of CPT-11.

## Figures and Tables

**Figure 1 molecules-28-01577-f001:**
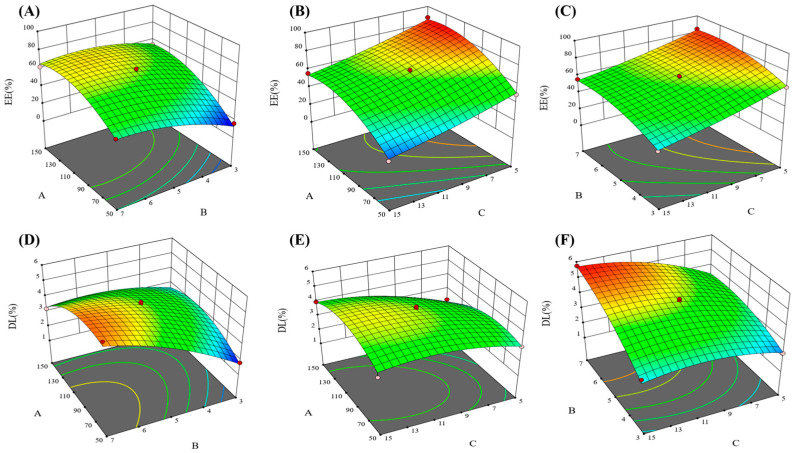
3D effect surface map of encapsulation efficiency and drug loading. In which (**A**–**C**) were encapsulation efficiency 3D effect surface map, and (**D**–**F**) were drug loading 3D effect surface map.

**Figure 2 molecules-28-01577-f002:**
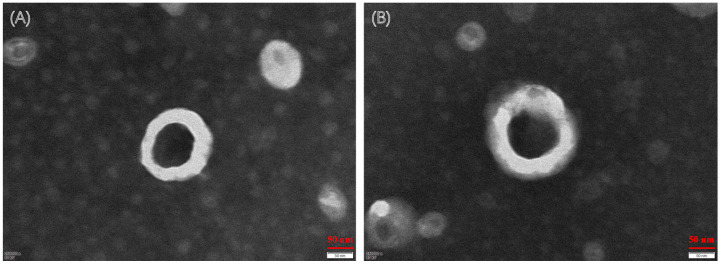
Transmission electron microscope (TEM) photographs of liposomes. TEM image of CPT-11-Lip (**A**). TEM image of CA-CPT-11-Lip (**B**).

**Figure 3 molecules-28-01577-f003:**
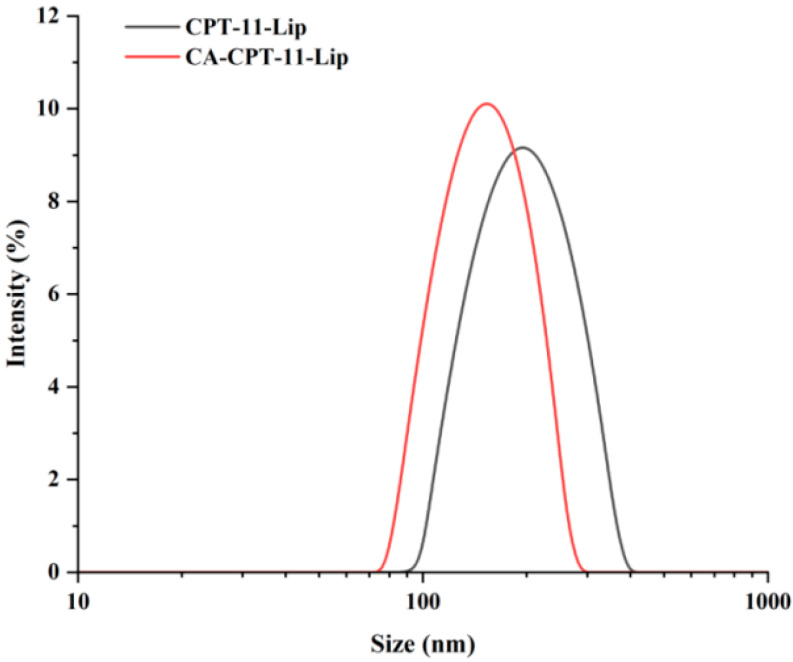
Liposome particle size diagram of CPT-11-Lip and CA-CPT-11-Lip.

**Figure 4 molecules-28-01577-f004:**
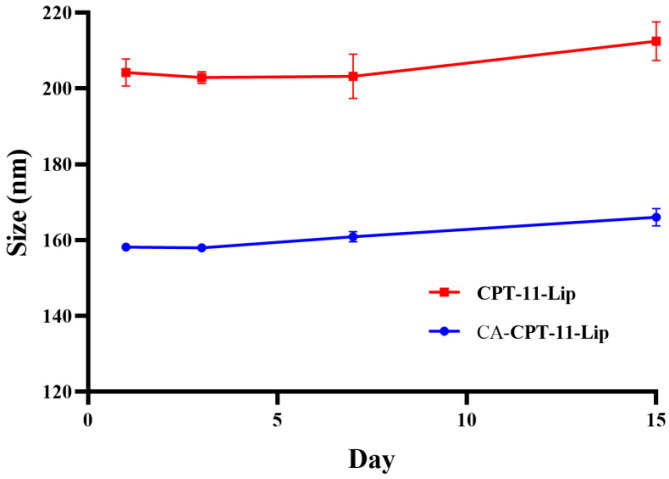
Changes in particle size of liposomes within 15 days.

**Figure 5 molecules-28-01577-f005:**
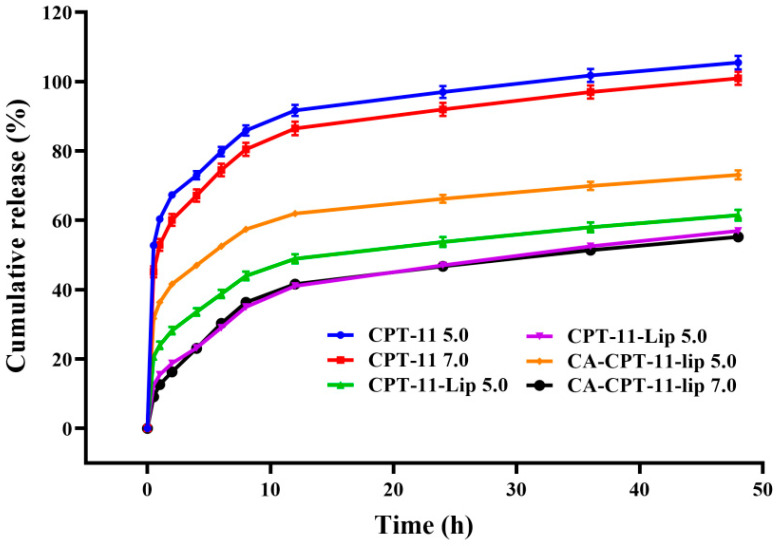
Cumulative release results of CPT-11, CPT-11-Lip, and CA-CPT-11-Lip in PBS solution of different pH.

**Figure 6 molecules-28-01577-f006:**
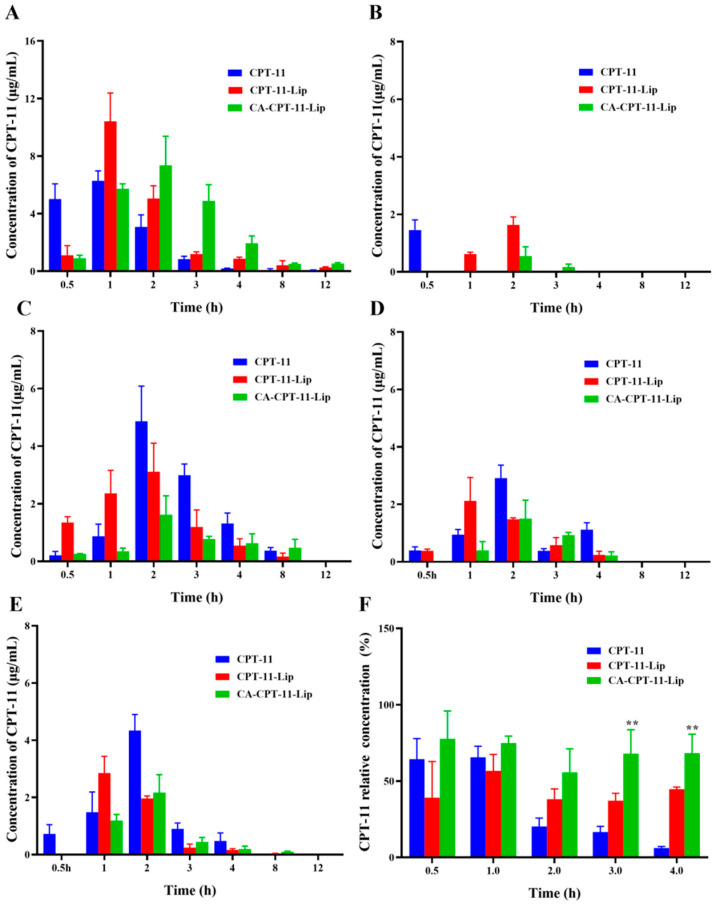
Concentration of CPT-11 in the liver (**A**), heart (**B**), spleen (**C**), lung (**D**), kidney (**E**), and the relative concentration of CPT-11 in the liver (**F**) (*n* = 3). Compared with the CPT-11-Lip group, *** p* < 0.01.

**Figure 7 molecules-28-01577-f007:**
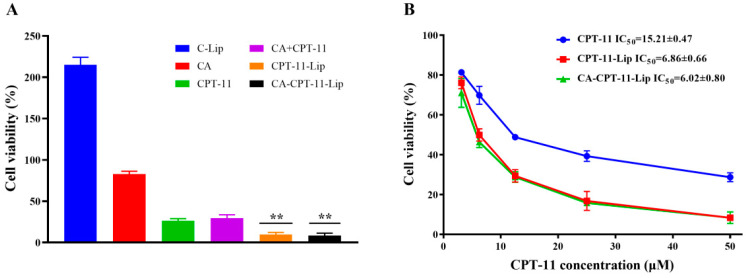
In vitro toxicity on HepG-2 cells of various preparations at 50 μM concentration (**A**) and in vitro HepG-2 cytotoxicity at different concentrations of CPT-11, CPT-11-Lip, and CA-CPT-11-Lip (**B**). Compared with CPT-11 group, *** p* < 0.01.

**Figure 8 molecules-28-01577-f008:**
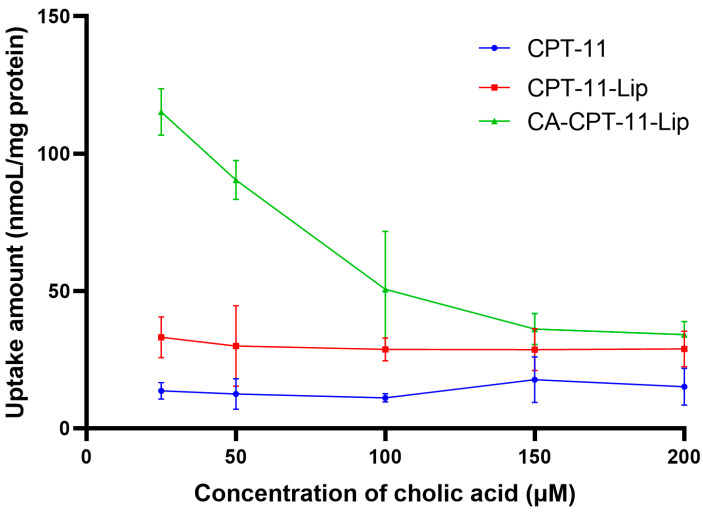
The effect of cholic acid on uptake of CPT-11 in HepG2 cell (*n* = 3).

**Table 1 molecules-28-01577-t001:** The design of factor levels.

Level	A Lecithin Dosage/mg	B The Ratio of Lecithin and Cholesterol	C CPT-11 Dosage/mg
−1	50	3:1	5
0	100	5:1	10
1	150	7:1	15

**Table 2 molecules-28-01577-t002:** BBD results.

NO.	A	B	C	EE (%)	DL (%)
1	0	1	−1	84.24	3.19
2	−1	1	0	42.59	5.47
3	0	0	0	62.31	4.38
4	1	−1	0	54.31	2.41
5	1	0	−1	87.38	2.20
6	0	0	0	61.47	4.23
7	−1	0	1	20.41	3.38
8	0	0	0	68.51	4.66
9	0	0	0	64.28	4.56
10	−1	0	−1	49.06	3.05
11	1	0	1	55.73	3.97
12	0	0	0	67.28	4.61
13	0	−1	1	32.66	2.99
14	1	1	0	61.71	3.20
15	0	1	1	55.78	5.75
16	0	−1	−1	62.94	2.25
17	−1	−1	0	16.06	1.80

**Table 3 molecules-28-01577-t003:** Entrapment efficiency ANOVA results.

Source	Sum of Squares	df	Mean Square	F-Value	*p*-Value
Model	5818.3	9	646.48	58.06	**
A-A	2145.45	1	2145.45	192.69	**
B-B	767.34	1	767.34	68.92	**
C-C	1771.32	1	1771.32	159.09	**
AB	91.49	1	91.49	8.22	*
AC	2.25	1	2.25	0.2021	0.6666
BC	0.8281	1	0.8281	0.0744	0.7929
A²	748.3	1	748.3	67.21	**
B²	241.36	1	241.36	21.68	**
C²	15.3	1	15.3	1.37	0.2794
Residual	77.94	7	11.13		
Lack of Fit	47.15	3	15.72	2.04	0.2507
Pure Error	30.79	4	7.7		
Cor Total	5896.24	16			

* *p* < 0.05, ** *p* < 0.01.

**Table 4 molecules-28-01577-t004:** Drug loading ANOVA results.

Source	Sum of Squares	df	Mean Square	F-Value	*p*-Value
Model	21.14	9	2.35	26.31	**
A-A	0.4608	1	0.4608	5.16	0.0573
B-B	8.32	1	8.32	93.21	**
C-C	3.64	1	3.64	40.82	**
AB	2.07	1	2.07	23.22	**
AC	0.5184	1	0.5184	5.81	*
BC	0.8281	1	0.8281	9.27	*
A²	2.91	1	2.91	32.6	**
B²	0.8022	1	0.8022	8.98	*
C²	1.08	1	1.08	12.1	*
Residual	0.6251	7	0.0893		
Lack of Fit	0.4972	3	0.1657	5.18	0.0729
Pure Error	0.1279	4	0.032		
Cor Total	21.77	16			

** p* < 0.05, *** p* < 0.01.

**Table 5 molecules-28-01577-t005:** Comparison of predicted and measured values.

	EE (%)	DL (%)
Sample 1	82.38	3.73
Sample 2	80.21	3.67
Sample 3	83.54	3.77
Average value	82.04	3.72
Predictive value	80.27	3.92
Deviation	1.77	−0.2

**Table 6 molecules-28-01577-t006:** Characterization of different liposomes (n = 3; mean ± SD).

	Size (nm) ± SD	PDI ± SD	ZP (mV) ± SD	EE (%)	LD (%)
CPT-11-Lip	197.70 ± 3.04	0.174 ± 0.038	−53.07 ± 1.47	74.54 ± 1.44	3.73 ± 0.09
CA-CPT-11-Lip	154.16 ± 4.92	0.146 ± 0.018	−56.93 ± 0.46	82.04 ± 1.38	3.72 ± 0.04

## Data Availability

The data presented in this study are available upon request from the corresponding author.

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
