# Peer review of "A “Trojan Horse” Strategy: The Preparation of Bile Acid-Modifying Irinotecan Hydrochloride Nanoliposomes for Liver-Targeted Anticancer Drug Delivery System Study"

_molecules, 2023, doi:10.3390/molecules28041577_

Round 1
Reviewer 1 Report
The article entitled “A “Trojan Horse” Strategy: The Preparation of Bile acid Modifying Irinotecan Hydrochloride nanoliposomes for liver targeted anticancer drug delivery System Study” submitted by authors is well-written with sufficient scientific data. The presentation is good; however, introduction need to be improved with more suitable existing examples of Trojan Horse Strategy with a picture or image for the cancer or any other related therapies in the literature.
The research problem is unique and impressive. The cholic acid-modified irinotecan hydrochloride liposomes (CA-CPT-11-Lip) strategy is great idea and authors able to achieve improvements such as smaller particle size, higher drug loading and encapsulation efficiency compared with the conventional liposomes (without cholic acid modification, named CPT-11-Lip). However, the drug release profile of both CA-CPT-11-Lip and CPT-11-Lip are similar without any significant improvement which is one of the key factors need to improve for the Trojan Horse strategy.
However, the research problem is very impressive and authors able to discovered the novel CA-CPT-11-Lip as a new Trojan Horse transporter with acceptable drug accumulation and release in liver.
After improving introduction part, I recommend this article to publish in Molecule Journal.
Reviewer 2 Report
Comments and Suggestions for Authors
The research titled “A “Trojan Horse” Strategy: The Preparation of Bile acid Modifing Irinotecan Hydrochloride nanoliposomes for liver-targeted anticancer drug delivery System Study” by Tao Zhou, and coworkers shed light on liver targeting using semi-synthetic anticancer drug, Irinitecan encapsulated into vesicular system thereafter they functionalized the drug delivery carrier by cholic acid based on approach of ‘Trojan horse’. Thereby, authors promised for efficient ligated carrier delivery into hepatocellular carcinoma. Further, the developed carrier system optimized by employing DoE, BBD and characterized in vitro for various parameters including drug loading, %EE, morphology, surface charge and, %DR, cytotoxicity and biodistribution. The drug targeting and specific delivery in cancer therapy including liver cancer through novel approach is most encouraging and topic of interest. I appreciate authors for conducting good work. Overall, after reviewing the manuscript, it was found suitable for publication with some concern that is being mentioned below. The written part of manuscript clear and adopted professional approach and however, data presented at some point very precise and unclear. At some points english/grammar, sentence consistency need to revise. From our side following points would be taken into consideration:
Comment 1: Enrich your introduction part with active targeting approach mediated via ligated nanocarrier.
See ref and cite; https://doi.org/10.1021/acsomega.1c06796
Comment 2: Authors estimated DL and EE using Sephadex G-50 gel chromatography method. Citation need.
Comment 3. Section 4.3. BBD study can further be elaborated for better understanding to the readers. See articles and cite; https://doi.org/10.3390/gels8040219
Comment 4: Simplify the equation (3) for %DR.
Comment 5: In TEM image appearance of vesicle and bar size is not visible.
Comment 6: In tissue distribution study, authors have not mentioned the no of animals in different groups. The tissue distribution study conducted for 12 hours only. %DR was observed for 48h, means after 12 h the concentration of drug still in the biological system or not. Why not distribution study measured after 12 h. Justification needed.
Comment 7. %DR study conducted at pH 5.0 and 7.0 why? The pH of biological system close to 7.4 and tumor microenvironment is slightly acidic.
Comment 8. Suggested to apply kinetic release model.
Comment 9. Suggested to the add cell uptake for better clarity on active targeting.
The manuscript could be considered for publication only after revision. Moreover, I am glad to review a revision of this manuscript if necessary.
Reviewer 3 Report
Although chemotherapy is still one of the main therapeutic approaches for treating liver cancer, not every patient is able to tolerate it due to the poor selectivity of the cytotoxic drugs used (which include irinotecan). In this connection, the development of new unconventional approaches using modern molecular-biological and biochemical methods to find new tumor-specific molecular targets and to understand the nature of this specificity in order to create more effective antitumor drugs is an urgent task.
There are no more significant remarks.
